# The Impact of Fixation on the Detection of Oligodendrocyte Precursor Cell Morphology and Vascular Associations

**DOI:** 10.3390/cells10061302

**Published:** 2021-05-24

**Authors:** Friederike Pfeiffer, Amin Sherafat, Akiko Nishiyama

**Affiliations:** 1Department of Physiology and Neurobiology, University of Connecticut, Storrs, CT 06269, USA; aminsherafat@gmail.com; 2Department of Neurophysiology, Institue of Physiology, Eberhard Karls University of Tübingen, 72074 Tübingen, Germany; 3Institute for Systems Genomics, University of Connecticut, Storrs, CT 06269, USA; 4The Connecticut Institute for Brain and Cognitive Sciences, University of Connecticut, Storrs, CT 06269, USA

**Keywords:** OPC, protrusion, PDGFRα, NG2, blood vessel, fixation, immunofluorescence

## Abstract

Oligodendrocyte precursor cells (OPCs) display numerous protrusions that extend into the surrounding parenchyma in the brain. Depending on the preparation of the tissue analyzed, these protrusions are more or less visible. We applied six different fixation methods and compared the effect of prolonged and stronger fixation on fluorescence intensity of platelet-derived growth factor receptor alpha, a surface marker of OPCs. Importantly, the fluorescence signal is mostly lost on protrusions as compared to the cell body, which has to be considered for specific analyses. Additionally, we show numerous contacts established between OPCs and the brain vasculature, which will contribute to the understanding of the interactions between these two elements.

## 1. Introduction

Oligodendrocyte precursor cells (OPCs) cells comprise a glial cell population that covers the entire parenchyma of the central nervous system (CNS). OPCs have the capacity for self-renewal throughout life. In addition to generating oligodendrocytes, some OPCs do not differentiate for months but remain as OPCs [1,2]. These non-vascular OPCs express the NG2 antigen and the alpha receptor for platelet-derived growth factor (PDGFRα), both of which are localized on the cell surface [3] and are commonly used to detect and visualize OPCs in immunohistochemical studies by antibodies against those proteins.

In order to study the functions of OPCs within the neural network, it is necessary to adequately label them for visualization. For many studies, visualizing the somata of OPCs is sufficient for the analysis of OPC numbers to assess proliferation or survival. However, OPCs also extend many protrusions into the surrounding brain parenchyma, which is why they are also referred to as “polydendrocytes” [2,4]. Investigations of the functions of OPCs require a detailed assessment of OPC morphology and localization, particularly at all the sites of contact they establish with other cells of the CNS both by light and electron microscopy. To this end, it is absolutely essential to optimize methods to visualize their numerous processes accurately and consistently at a high resolution. The quality of immunodetection largely depends on how the tissue is prepared.

Anecdotally, it is widely recognized that extensive tissue fixation by aldehydes preserves tissue morphology but compromises antigen detectability due to epitope masking [5,6]. So far, there is one report focusing on OPC proliferation in the cortex, which describes the effect of the concentration of formaldehyde used to fix the tissue on the quality of subsequent immunohistochemistry [7]. The authors demonstrated that, specifically, the staining for NG2 was greatly reduced with a higher concentration of formaldehyde. Higher concentrations of formaldehyde thus led to an underestimation of the number of NG2-expressing OPCs in the respective brain area. By contrast, lower concentrations resulted in dramatically improved immunostaining for NG2 in adult mouse brains.

The standard fixation for light microscopy is 4% paraformaldehyde (PFA) perfusion followed by overnight post-fixation. Identifying sites of contacts between different cell types often requires immunolabeling of specific cellular markers in order to identify cell types. In this case, a reasonable compromise has to be made between strong fixation resulting in suitable tissue preservation and weaker fixation enabling antibody diffusion and antigen recognition within the fixed tissue. The addition of 0.1 M L-lysine and 0.01 M sodium metaperiodate to 4% PFA (PLP fixation) has been shown to enhance the cross-linking of glycoproteins at the cell surface [8]. Since the two integral membrane glycoproteins PDGFRα and NG2 are commonly used to detect OPCs, we have been routinely using PLP fixation in our immunolabeling studies with satisfactory results.

For pre-embedding immunoelectron microscopy, 0.1–0.5% glutaraldehyde (GA) is often added to PFA for enhanced preservation of the ultrastructure. In this study, we compared the effects of 1 h or overnight (O/N) PFA fixation on PDGFRα immunofluorescence intensity and the ability to visualize detailed OPC morphology, particularly the preservation of PDGFRα on distal OPC processes at the site of their contact with blood vessels. We further evaluated the effects of additional 0.1% and 0.5% GA on OPC immunofluorescence intensity. Collectively, our findings revealed beneficial effects of short fixation on the preservation of the PDGFRα antigen, even in the presence of GA, without compromising tissue integrity.

## 2. Materials and Methods

### 2.1. Animals

All animal procedures were approved by the Institutional Animal Care and Use Committee of the University of Connecticut. Wild-type C57BL/6J mice were obtained from Jackson Laboratories (Stock#000664). We examined two age groups, young mice at P15 and adult mice at P60 of both sexes.

### 2.2. Tissue Processing

Mice of the two age groups were perfused and fixed with the following solutions:

The standard fixing solution used in our experiments was 4% PFA containing 0.1 M L-lysine and 0.01 M sodium metaperiodate (PLP) [8]. After perfusion, animals were dissected and brains post-fixed in PLP fixative for either 1–2 h or overnight (O/N) and then washed 4 times in 0.2 M sodium phosphate buffer, pH 7.4. The brains were incubated in 30% sucrose in 0.2 M sodium phosphate buffer for at least 48 h and then frozen in OCT compound (Tissue-Tek; Adwin Scientific 14-373-65). Sections of 15-µm were cut on a Leica CM3050S cryostat.

Alternatively, mice were perfused with either 4% PFA only or 4%PFA containing either 0.1% or 0.5% GA (Electron Microscopy Sciences), dissected, and brains post-fixed in the same fixative for 1 h. The tissue was further processed as described above. The different fixation conditions are summarized in Table 1.

### 2.3. Immunohistochemistry

For immunolabeling, sections were rinsed in PBS, blocked in 1% bovine serum albumin (BSA) containing 0.3% Triton X-100 in PBS for 1 h at room temperature, followed by incubation with primary antibodies (rabbit anti-laminin, Biomedical Technologies BT-594, diluted 1:1000; goat anti-PDGFRα, R&D Systems AF1062, diluted 1:500, rabbit anti-NG2, Millipore AB5320, diluted 1:500) in 1% BSA/0.1% Triton X-100 in PBS overnight. The sections were rinsed 3 times in PBS and incubated in the secondary antibodies (Cy3-donkey anti-rabbit #711-165-152 and Alexa488 bovine anti-goat #805-545-180, both Jackson Research) diluted 1:500 in 1%BSA/0.1%Triton X-100 in PBS at room temperature for 1 h. Sections were rinsed and mounted with Vectashield containing DAPI (Vector, H-1200).

### 2.4. Fluorescence Microscopy

Images of fluorescently labeled tissue sections were acquired using Leica SP8 confocal microscope from the neocortex and corpus callosum (Bregma −1.46 to −2.15 mm). For each sample, successive images were taken from cortical layers 1–3, layers 4/5, and layer 6/corpus callosum. Images were analyzed with the Leica LAS X (for 3D reconstruction) or ImageJ2 (Fiji). All images were acquired with the same laser intensity, gain, and resolution, 40 *z*-stacks per image at 0.35 µm *z*-step size.

### 2.5. Quantification of Fluorescence Intensity and Data Analysis

Image J2 was used for measuring fluorescence intensity. The channel displaying the PDGFRα-signal was processed as follows: the values from a constant number of 40 *z*-stacks in an area of 52,007.876 square pixels, was combined (sum slices), background was subtracted (rolling ball radius of 50 pixels), and the ‘mean gray value’ for the entire field in each image was measured. The value for the control image from the same sample, where the primary antibody had been omitted, was subtracted. The resulting values were pooled together for every experimental group and cortical layer group (n = 4) and subjected to statistical analysis. GraphPad Prism 9 was used to perform two-way ANOVA followed by Tukey’s multiple comparisons test (to compare the same layer of each fixation protocol applied). For the P60 groups, measurements were taken from a defined area of 2345.876 square pixels around the cells through all the confocal stacks.

In order to visualize contacts between OPC protrusions and the vasculature, single *z*-slices were selected and used after subtracting the background as described above, and a maximum projection of 40 stacks is provided as an overview.

## 3. Results

The quality and efficiency of tissue fixation critically determine the outcome of the immunohistochemical procedure and, ultimately, the quality of the resulting images, which form the critical basis for correct interpretation of biological phenomena. Therefore, we applied several fixation protocols and compared the quality of OPC staining thereafter.

### 3.1. Choice of Fixatives

Commonly, 4% PFA is used as a fixing solution for light microscopy. To achieve better detection of the cell surface markers for OPCs, we have been routinely using 4% PFA containing 0.1 M L-lysine and 0.01 M sodium metaperiodate (PLP), which is suitable for detecting cell surface glycoprotein antigens, including NG2 [9] and PDGFRα [10]. We typically post-fix the brain tissue for 1–2 h after perfusion, but as many commonly used protocols use O/N fixation with 4%PFA, we compared our short fixation protocol with an O/N fixation protocol for both fixatives, 4%PFA and PLP at P15. Additionally, we compare short and O/N fixation with PLP in two age groups (P15 PLP 1 h vs. P15 PLP O/N and P60 PLP 1 h vs. P60 PLP O/N) (Table 1). Since it is generally recommended to add small amounts of GA to the fixative for immunoelectron microscopy in order to increase tissue preservation, we also analyzed the quality and intensity of PDGFRα staining after adding either 0.1%GA or 0.5%GA to the base 4%PFA in the young age group (P15 0.1%GA 1 h and P15 0.5%GA 1 h) and compared to P15 mice fixed with 4% PFA only (P15 4%PFA 1 h and P15 4%PFA O/N) or PLP fixative without GA (P15 PLP 1 h and P15 PLP O/N) (Table 1, P15 group).

### 3.2. Effects of Fixation on PDGFRα Fluorescence Intensity and Morphological Detail

In order to determine the impact of different fixation protocols on OPC morphology and cell surface antigen detection, we compared the intensity and quality of PDGFRα immunofluorescence in P15 and adult brain tissue fixed under different conditions (Table 1). Since PDGFRα expression levels diminish beyond the second postnatal week (Nishiyama, 1996), we examined the effect of fixation on brain tissue from P15 and P60 mice for samples with different levels of PDGFRα expression on OPCs. We measured PDGFRα immunofluorescence intensity in three different depths within the dorsal telencephalon: (1) outer layers (cortical layers 1–3; L1–3), middle layers (cortical layers 4–5; L4–5), and deep layers (cortical layer 6 and corpus callosum; L6-CC). We reasoned that the efficacy of tissue penetration of the fixative during post-fixation and the extent of solution exchanges during the washes might be affected as a function of the distance from the pial surface, as well as the density of the tissue that increases with age.

#### 3.2.1. Effect of Increased Fixation on Fluorescence Intensity of PDGFRα Signal in P15 Mice

We first compared all six fixation conditions applied to P15 mice. Throughout all layers of the dorsal telencephalon, the overall fluorescence intensity appeared highest in the P15 4%PFA 1 h group (Figure 1A,G,M). In the outer cortical layer 1–3, the fluorescence intensity for the 4%PFA and PLP short fixation protocols was similarly high and not significantly different (Figure 1A,C). In the deeper layers 4–5 and 6-corpus callosum, the fluorescence intensity for short fixation with 4%PFA was significantly higher compared to short fixation with PLP (Figure 1G,M compared to Figure 1I,O). By prolonging the 4%PFA (Figure 1B,H,N) or PLP (Figure 1D,J,P) post-fixation period to O/N, fluorescence intensity decreased 1.7 to 2.1-fold in all layers for the 4%PFA fixation protocol, and 2.2- to 2.4-fold in all layers similarly within the P15 PLP O/N group (Figure 2, compared 1 h to O/N). After 4% PFA or PLP O/N fixation, the PDGFRα-signal was still detected in the soma (arrows in Figure 1B,H,N,D,J,P). The number of detectable OPCs was not affected by prolonging the post-fixation period (Appendix A). The extended fixation time caused a more significant loss of the signal from OPC protrusions, resulting in fewer visible protrusions, and those that remained appeared shorter and less numerous. This tendency appeared to be more pronounced in 4% PFA 1 h group compared with the PLP 1 h group. The appearance of OPCs and their protrusions was best preserved in the PLP 1 h fixation group (Figure 1C,I,O), despite slightly lower fluorescence as compared to the 4%PFA 1 h fixation group. Thus, we conclude that for the analysis of fine structures and morphology, the PLP 1 h protocol is most suitable.

When we added 0.1% GA to the fixative for perfusion and post-fixed for 1 h with 0.1%GA/4%PFA and compared the outcome with the P15 4%PFA 1 h group, there was a clear reduction in fluorescence signal as well, but not a uniform loss among the layers. In the outermost layers L1–3, PDGFRα immunofluorescence intensity was reduced 3.6-fold to a level even lower than that of the PLP O/N group (Figure 1A vs. Figure 1E and Figure 2 left group). In the inner layers (L4–5 and L6-CC) addition of 0.1%GA significantly reduced PDGFRα immunofluorescence intensity compared with the P15 4%PFA 1 h group (Figure 1G vs. Figure 1K,M vs. Figure 1Q and Figure 2 middle and right groups), but the differences were not significant when compared to the P15 PLP 1 h group (Figure 1I vs. Figure 1K,O vs. Figure 1Q). Thus, the loss of signal was most pronounced in L1–3, while antigenicity was better preserved in the inner layers of L4-CC.

A similar layer-dependent effect was observed when we added 0.5% GA to the base 4% PFA fixative. While PDGFRα immunofluorescence intensity was significantly reduced across all layers compared with the P15 4%PFA 1 h group, the signal intensity was most dramatically decreased in the outer layers, 4.6-fold, in the outermost layer (L1–3; Figure 1A vs. Figure 1F and Figure 2, left). A total of 0.5% GA reduced PDGFRα immunofluorescence even more in the middle layer (L4–5) by 5.7-fold compared with P15 4%PFA 1 h (Figure 1G vs. Figure 1L and Figure 2, middle). The effect was only 2.7-fold in the innermost layer L6-CC due to the stronger signal in the 0.5% GA 1 h group in this layer (Figure 1M vs. Figure 1R and Figure 2, right).

Surprisingly, the fluorescence intensity for P15 0.1%GA 1 h in the middle layers L4–5 was higher than the value for P15 PLP O/N. In the inner layers L6-CC, the values for both GA groups were higher compared to the P15 PLP O/N group and the value for the P15 0.1%GA 1 h group was even slightly higher compared to the value for the P15 4%PFA O/N group, indicating the greater deleterious effects of prolonged fixation as compared to adding GA in the inner layers. Of note, while the cell bodies of PDGFRα-expressing OPCs had the most intense signal and were visible in all groups, the number of protrusions observed where greatly diminished in all protocols except the short 4%PFA and PLP fixation protocols (Figure 1A,G,M and Figure 1C,I,O, respectively), and thereby the differences between the protocols were most pronounced in the outermost layers L1–3 (Figure 2, L1–3).

#### 3.2.2. Effect of Increased Fixation on Fluorescence Intensity of PDGFRα Signal in P60 Mice

When we examined older animals (P60, Figure 3 and Figure 4), we saw a drop in the percentage of OPCs in the motor cortex from 7% at P15 to 3% at P60 (Appendix A), consistent with previous reports on the lower density of OPCs in the adult CNS parenchyma (5–9% of cells in adult rats [11] compared with 20% OPCs in the white matter of P10 animals [12,13]). As in P15 mice, the number of OPCs was not affected by the prolonged fixation (Appendix A).

Since it has been shown that PDGFRα levels decline in the more mature brain [3], we examined the effects of fixation on PDGFRα immunofluorescence to determine whether the ability to detect PDGFRα was more susceptible to overfixation. As predicted, the overall level of PDGFRα immunofluorescence intensity was significantly lower in P60 brains than in P15 brains (Figure 1 and Figure 3) and was further reduced by prolonging the post-fixation period from 1 h to overnight with PLP, as we had observed for the P15 animals. The average fluorescence intensity per image was 2.1-fold lower in P60 PLP 1 h as compared to P15 PLP 1 h hour in L1–3, 2.3- fold lower in L4–5, and 3.0-fold lower in L6-CC. When comparing P60 PLP O/N to P15 PLP O/N, the fluorescence intensity was almost the same in L1–3 (only 1.03-fold reduced), in L4–5 it was 1.4-fold reduced, and again the most striking significance was observed in L6-CC, where the fluorescence intensity was reduced 2.8-fold when comparing both age groups. At P60, the fluorescence intensity of PDGFRα labeling was 1.7- to 2.2-fold lower in the O/N group compared to the 1 h fixation group, with the difference being significant in all layers (compare Figure 3A vs. Figure 3B, Figure 3C vs. Figure 3D, Figure 3E vs. Figure 3F and Figure 4). Thus, the trend of losing fluorescence intensity with prolonged PLP fixation remained at P60 through all the layers. As in the P15 experimental groups, prolonged fixation more severely reduced the detection of PDGFRα immunolabeling on OPC protrusions compared to cell bodies.

### 3.3. Relationship between OPC Processes and Vascular Elements

Having described the difference in fluorescence intensity between the different fixation groups, we proceeded to examine OPC protrusions contacting other cell types of the CNS. We focused on blood vessels visualized with an antibody to laminin, which is a component of the basement membrane around the majority of the blood vessels in the brain [14]. Contacts between OPCs and vessels have been observed previously [15,16]. We compared the spatial relationship between PDGFRα+ OPC processes and laminin-positive blood vessels in the P15 PLP 1 h and P15 PLP O/N groups. In P15 PLP 1 h samples, we were able to observe the close proximity between OPC protrusions and laminin-positive blood vessels (Figure 5A–D, shown as 3D-rendered images of 40 confocal *z*-stacks). However, this close proximity of protrusions at the vasculature was less extensive in the O/N fixed brain tissues (Figure 5E–H).

We further examined the spatial relationship between PDGFRα-positive protrusions of OPCs and laminin-positive vessels in the P15 mouse brain by analyzing serial single *z*-slices, each spanning 0.35 μm. Figure 6 shows two OPCs, each of which was contacting two distinct vessels with two of its protrusions (protrusions of OPC1 are marked with arrowheads, protrusions of OPC2 are marked with arrows). OPC1 extended one process toward a vessel above it (vessel a) and another one toward the vessel to the right (vessel b). Vessel b was also contacted by OPC2, which extended additional processes, one of which contacted the vessel at the top (vessel c). In another field, an OPC had its cell body adjacent to the vessel, with some of its protrusions extending along the vessel (OPC1 and arrows in Appendix A), while other OPCs extended protrusions toward a more distant vessel that their soma was not in contact with (OPC2 in Appendix A and arrowheads in Appendix A). Some OPCs extended several protrusions toward the same vessel (arrow and arrowhead in Appendix A). We also observed an OPC that was directly adjacent to a curved laminin-positive vessel (Appendix A), projecting multiple protrusions along the same vessel they were associated with (arrows in Appendix A). The associations between OPCs and blood vessels could be detected at blood vessels of different sizes, from capillaries to arterioles. These details could not be obtained in P15 PFA O/N samples. Thus, with optimum fixation protocol, we were able to observe a variety of contacts between OPCs and blood vessels in the P15 1 h group that we would have missed by applying prolonged or stronger fixation protocols.

To further evaluate the relationship between OPC processes and blood vessels, we performed a quantitative analysis of the contacts between OPC and blood vessels and how this analysis was affected by the different fixation protocols. When we compared the percentage of OPCs that were contacting blood vessels among age groups and fixation protocols, we found the following: in the P15 1 h group, 81% of OPCs were found to extend protrusions that were in contact with blood vessels, which decreased to 47% in the P15 O/N group (Figure 7). In the P60 1 h group, the percentage of OPCs contacting blood vessels had even increased to 94%, while at P60 O/N, only 49% of OPCs were detected to contact blood vessels (Figure 7). Conversely, when we examined the proportion of blood vessel segments that were contacted by an OPC protrusion, we found that in the P15 1 h group, 92% of blood vessels were contacted by OPC protrusions, whereas in the P15 O/N group, only 58% of blood vessels had detectable contact with OPCs (Figure 8). At P60, 83% of blood vessels were detected to have contacts established with OPCs after 1 h post-fixation, while only 47% of blood vessels were detected as being contacted by OPCs after O/N fixation (Figure 8).

In line with these observations, when we counted the blood vessels that did not show obvious contact to OPC protrusions, we found that only 8% of blood vessel segments per image did not have contact to OPCs in the P15 1 h, and this number increased to 41% with O/N fixation. At P60, 18% of blood vessel segments did not show contacts with OPCs and their protrusions after 1 h post-fixation, which increased to 55% with O/N fixation (Figure 9). Thus, overfixation significantly reduced our ability to detect fine OPC-vascular contacts.

### 3.4. PDGFRα Versus NG2 Staining

Finally, we compared the staining pattern of the two surface molecules generally used to characterize OPCs, namely NG2 and PDGFRα, in P15 PLP 1 h versus P15 PLP O/N groups (Figure 10). Both antigens were detected on OPCs as expected in P15 1 h group (Figure 10C). We noted slight differences in their staining pattern. While anti-PDGFRα staining was prominent on OPC bodies and was also detected on their numerous protrusions (Figure 10A), staining for NG2 resulted in less pronounced staining of the cell body but labeled more distal protrusions as compared to PDGFRα (Figure 10B). We have also noted that PDGFRα immunoreactivity varies with different antibodies and the weaker antibodies labeled soma but did not reveal the processes as much as the stronger antibodies (not shown).

As previously reported, anti-NG2 antibody stained pericytes on blood vessels that were PDGFRα-negative (arrows in Figure 10). We also observed very rare NG2+ glial cells that were negative for PDGFRα, which may be immature oligodendrocytes that had down-regulated PDGFRα (arrowhead in Figure 10A–C). Prolonged fixation with PLP in the P15 PLP O/N group greatly decreased the number of protrusions detected with both antibodies (Figure 10F) when compared to the P15 1 h group. As already observed in the P15 animals, staining for PDGFRα (Figure 10D) was more prominent on cell bodies, while staining for NG2 (Figure 10E) revealed more protrusions per OPC. Thus, strong fixation of brain tissue resulted in the loss of morphological information in brain tissue of young mice when staining with both commonly used markers for OPCs, NG2 itself, and PDGFRα.

In summary, these findings indicate that prolonged O/N fixation protocols commonly used for CNS tissue resulted in a severe reduction in the ability to detect morphological details by immunohistochemical staining with antibodies directed against OPC surface molecules. Of note, while OPCs themselves remained detectable, information about the existence and localization of their processes was greatly reduced, which should be considered when analyzing interactions between different cell types in the CNS. In our hands, a post-fixation period of 1–2 h using the fixative PLP was enough to ensure suitable tissue preservation for the purpose of light microscopy, while the ability to discern morphological details greatly benefited from the increased retention of the OPC surface antigens with shorter fixation.

## 4. Discussion

Morphological details are crucial when analyzing the interactions between different cell types in the CNS. In this study, we have shown the impact of the fixation protocol applied on the outcome of the immunohistochemical detection of surface markers NG2 and PDGFRα on OPCs. Prolonged post-fixation with 4%PFA or the addition of small amounts of GA decreased PDGFRα immunofluorescence signal, resulting in a reduced number of protrusions visible per OPC, although the OPC cell bodies were still detectable. We performed a side-by-side comparison of commonly used fixatives. The effect of stronger fixation was not only limited to detecting PDGFRα but also significantly reduced detection of NG2, another OPC surface glycoprotein antigen. The compromised immunofluorescence caused by prolonged fixation significantly diminished our ability to detect the association of distal OPC protrusions with the vasculature.

### 4.1. Effects of the Composition of the Fixative on PDGFRα Detection

PFA and GA are routinely used for tissue preparation. Compared to PFA, GA is a dialdehyde containing two reactive groups and is therefore very efficient in inter- and intramolecular binding, resulting in highly cross-linked proteins and suitable preservation of the ultrastructure and tissue integrity. This comes at the expense of antigenicity, as many antigens will be masked or not accessible to the antibodies used.

We routinely fix rodent brains for 1–2 h using the PLP fixative, which contains L-lysine and sodium metaperiodate in addition to PFA, as a standard for perfusion and post-fixation for detecting OPC surface antigens as well as cytoplasmic and nuclear antigens. This fixative stabilizes carbohydrate residues and is therefore suitable for detecting cell surface glycoprotein antigens. While PFA generally stabilizes proteins and lipids, the sodium metaperiodate (NaIO_4_) additionally oxidizes carbohydrate residues, thereby generating aldehyde groups that can also be cross-linked by PFA. Additionally, lysine cross-links the carbohydrate-containing molecules by reacting with the aldehyde groups. Using this fixative provides the advantage of preserving antigenicity to the same extent as PFA. PLP fixation has also been shown to preserve the ultrastructure comparable to glutaraldehyde, which is commonly used in electron microscopy [8] and has been proven to be highly suitable for the staining of surface glycoprotein antigens on OPCs, enabling visualization OPC protrusions while also preserving tissue integrity. By direct comparison between the two fixatives, 4%PFA and PLP, we can show that although using 4%PFA results in higher fluorescence intensity, PLP is superior in preserving fine structural details on OPCs and their numerous protrusions (Figure 1). As GA is highly efficient in cross-linking lysine, adding L-lysine and sodium metaperiodate to the fixing solutions containing GA caused premature cross-linking and precluded us from adding them to GA-containing fixatives.

### 4.2. Effects of the Duration of Fixation

As many laboratories use protocols with O/N duration of the post-fixation period, we compared a 4%PFA 1 h hour post-fixation protocol with a 4% PFA O/N fixation to our routine procedure, which involves perfusion with PLP followed by a short 1 h post-fixation in the same fixative and PLP perfusion followed by O/N post-fixation. In addition, we evaluated the effects of two fixatives containing different amounts of GA on the intensity of PDGFRα immunofluorescence signal in P15 mice. While prolonging the post-fixation period with 4%PFA or PLP to overnight similarly reduced the intensity of the PDGFRα immunofluorescence signal in all layers of the dorsal telencephalon analyzed, the addition of 0.1% GA to PFA most significantly decreased signal in the superficial layers (L1–3), while the signal was better preserved in the inner layers toward the corpus callosum (L4–6 and CC). This may be a result of the higher cross-linking capability of GA as compared to PFA, resulting in a decreased penetration depth of the fixative during the post-fixation period. This would mean that epitope masking could be most pronounced in the superficial layers due to the accessibility of GA to the tissue, an important factor to be considered when sampling tissues for immunoelectron microscopy.

The differences in fluorescence intensity after applying various fixation methods were similar in young mice at P15 compared with adult mice at P60. However, because of the lower level of PDGFRα receptors in P60 brain compared with younger brains, as previously reported [13], the diminished immunofluorescence caused by prolonged fixation renders PDGFRα barely detectable on OPC protrusions. These observations underscore the importance of optimized fixation for cell surface antigens, particularly when they are less abundant.

The observations that prolonged fixation or addition of GA to the PFA-based fixative reduced NG2 immunofluorescence is consistent with the previous report that lighter O/N fixation with lower formaldehyde concentrations yielded superior NG2 immunofluorescence [7]. Our findings indicate that tissue integrity, as judged by the preservation of OPC process arborization, is not compromised with short fixation in the presence of 4% PFA. Thus, shortening the post-fixation time might provide the advantage that it can be used with a higher concentration of aldehydes to achieve better tissue preservation.

### 4.3. Effects of Fixation on the Ability to Detect OPC-Vascular Contacts

Contacts between OPCs and other structures such as blood vessels would be underestimated or even be dismissed if suboptimum tissue processing compromised the ability to visualize distal OPC processes. As an example of studying how OPCs interact with other cell types of the CNS, we focused on OPC protrusions contacting blood vessels and how the detection of such contacts was affected by fixation, a process that is easily overlooked when the tissue is over-fixed, and the protrusions are not clearly visible. We have found that OPCs established multiple contacts with the vasculature in different ways, either with vessels adjacent to their cell bodies or vessels several µm away via OPC protrusions. Thus, OPCs contacted the same vessel segment several times or contacted different vessel segments localized on opposite sides of their cell bodies. Our findings demonstrate a great reduction in the proportion of OPC-blood vessel contacts that are detected after prolonged O/N fixation.

The neurovascular unit (NVU), regulating the blood flow in the CNS, is not only composed of vascular cells such as endothelial cells, pericytes, and smooth muscle cells but also glial cells and neurons are regarded as being part of the NVU [17,18]. Astrocytes, with their end-feet tiling and covering the CNS vasculature, are the most studied cell type among the glial cells with regard to blood-brain barrier regulation. Contacts between different cell types of the CNS are relevant in neurovascular coupling, where neuronal activity is linked to blood flow and nutrient supply for active regions. Astrocytes have been long suggested to be involved in neurovascular coupling (NVC), where local neuronal activity can increase the local blood flow [19,20].

Although an earlier study revealed a paucity of OPC contacts at blood vessels compared to extensive vascular apposition by developing astrocytes [21], there is growing evidence from recent studies suggesting a more significant interaction between OPCs and the vasculature. Cerebrovascular endothelial cells secrete factors that promote OPC survival and proliferation via Akt activation, suggesting the presence of an “oligovascular niche” [22]. OPCs also proliferate and associate with microvessels during experimental autoimmune encephalomyelitis (EAE) [23]. A close association to the vasculature during development was proposed to serve as a guidance substrate for the migration of OPCs to the respective areas of the CNS, possibly coordinated with differentiation [16]. This may continue in the adult CNS, where not only perivascularly located OPCs but those whose somata are located away from vessels extend their protrusions toward pericytes [24], and secreted molecules from pericytes and OPCs appear to mutually influence the number of the other cell type. Conversely, OPCs influence vascular development in the developing embryo [25]. In pathological conditions, OPCs modify blood-brain barrier integrity [26,27,28]. Taken together, there is evidence arising for a functional interplay between vascular cells and OPCs.

One functional consequence of the close interaction between OPC protrusions and blood vessels could be the regulation of oligodendrocyte development in the context of the neural network. It has recently been shown that neuronal activity can regulate CNS myelination via the vasculature. Reduction in neuronal activity leads to a decrease in vascular endothelin expression, which in turn decreases myelin sheaths produced per oligodendrocytes [29]. One can imagine that the close proximity between OPCs and vessels would facilitate such a regulative mechanism to generate the correct amount of myelin to match the level of neuronal activity.

## 5. Conclusions

In summary, our findings suggest that OPCs are an integral component of the neurovascular unit. Our study provides the morphological basis of OPC association with the vasculature, providing clues as to how OPCs closely integrate signals from the complex vascular environment. The ability to accurately detect fine structures of OPCs will greatly facilitate the correct interpretation of the nature of the cell-cell interactions in the CNS. This, in turn, will impact the analysis of specific functions of OPCs, especially in the context of their interaction with other cells of the CNS.

## Figures and Tables

**Figure 1 cells-10-01302-f001:**
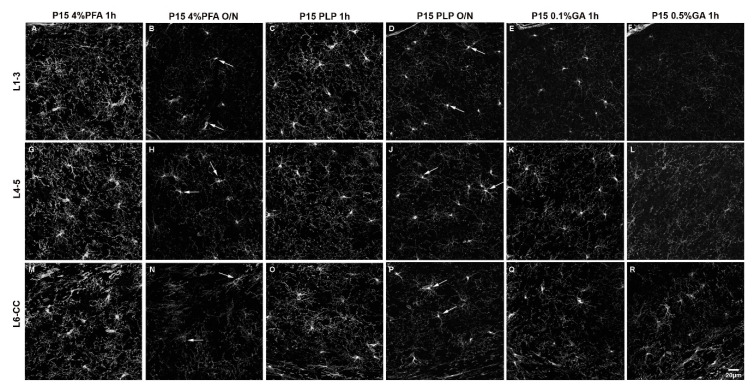
The effects of different fixation protocols on PDGFRα immunofluorescence at P15. (**A**–**F**): cortical layers 1–3, (**G**–**L**): cortical layers 4/5, (**M**–**R**): cortical layer 6/corpus callosum. P15 4%PFA 1 h (**A**,**G**,**M**) results in the strongest signal, comparable to P15 PLP 1 h (**C**,**I**,**O**). In P15 4%PFA O/N (**B**,**H**,**N**), P15 PLP O/N (**D**,**J**,**P**), P15 0.1%GA 1 h (**E**,**K**,**Q**) or P15 0.5%GA 1 h (**F**,**L**,**R**) the PDGFRα signal is decreased. Arrows indicate PDGFRα-positive OPC cell bodies that are still detectable after O/N fixation. Scale bar represents 20 µm.

**Figure 2 cells-10-01302-f002:**
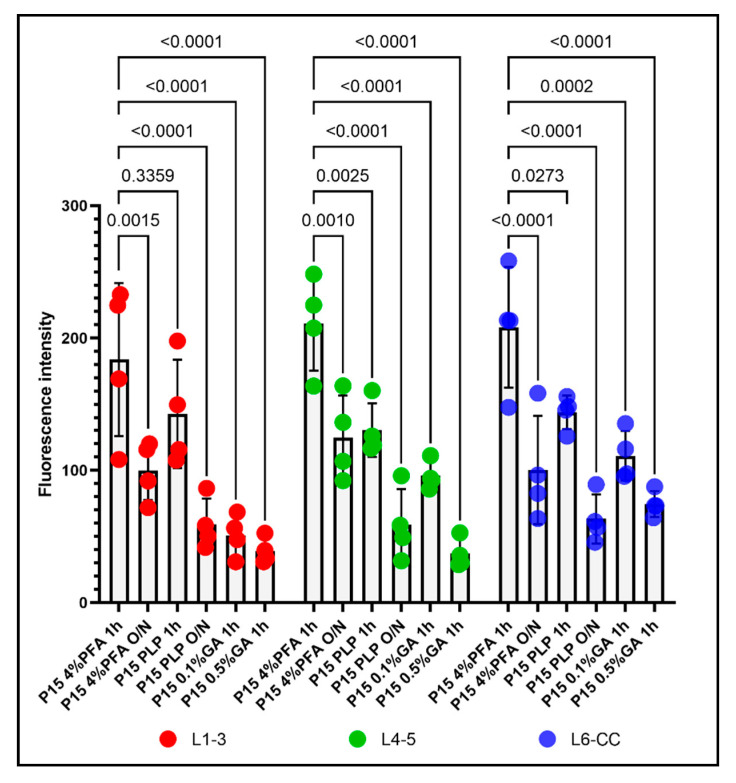
The effects of the different fixation protocols on PDGFRα immunofluorescence intensities at P15, grouped by different layers. Left: PDGFRα immunofluorescence intensity in L1–3 after 4%PFA 1 h, 4%PFA O/N, PLP 1 h, PLP O/N, 0.1%GA 1 h, and 0.5%GA 1 h. Middle: PDGFRα immunofluorescence intensity in L4–5 after 4%PFA 1 h, 4%PFA O/N, PLP 1 h, PLP O/N, 0.1%GA 1 h, and 0.5%GA 1 h. Right: PDGFRα immunofluorescence intensity in L6-CC after 4%PFA 1 h, 4%PFA O/N, PLP 1 h, PLP O/N, 0.1%GA 1 h, and 0.5%GA 1 h. Two-way ANOVA followed by Tukey’s multiple comparisons test. *n* = 4 for each group. Comparison among fixatives: F(5, 54) = 46.37, *p* < 0.0001. Comparison among layers: F(2, 54) = 3.350, *p* = 0.0425.

**Figure 3 cells-10-01302-f003:**
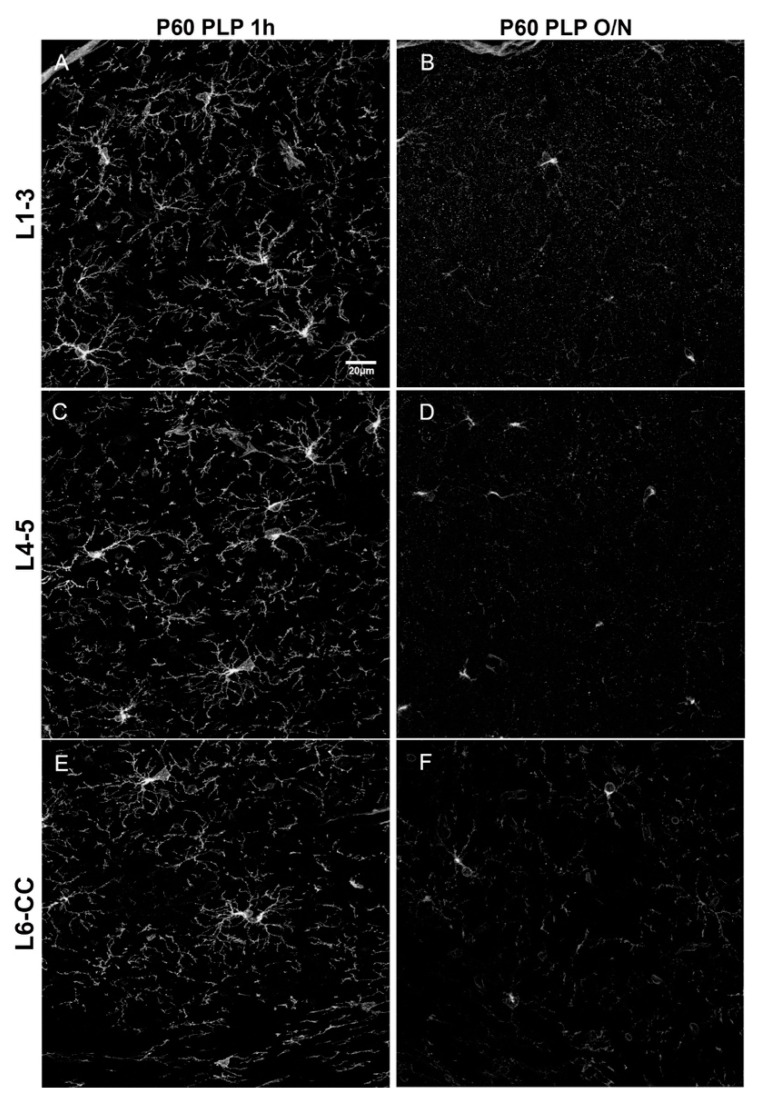
Comparison of different fixation protocols at P60. (**A**,**B**): cortical layer 1–3, (**C**,**D**): cortical layer 4/5, (**E**,**F**): cortical layer 6/corpus callosum. OPCs are detected by anti-PDGFRα antibody. P60 PLP 1 h (**A**,**C**,**E**) results in a stronger PDGFRα signal as compared to P60 PLP O/N (**B**,**D**,**F**). Scale bar represents 20 µm.

**Figure 4 cells-10-01302-f004:**
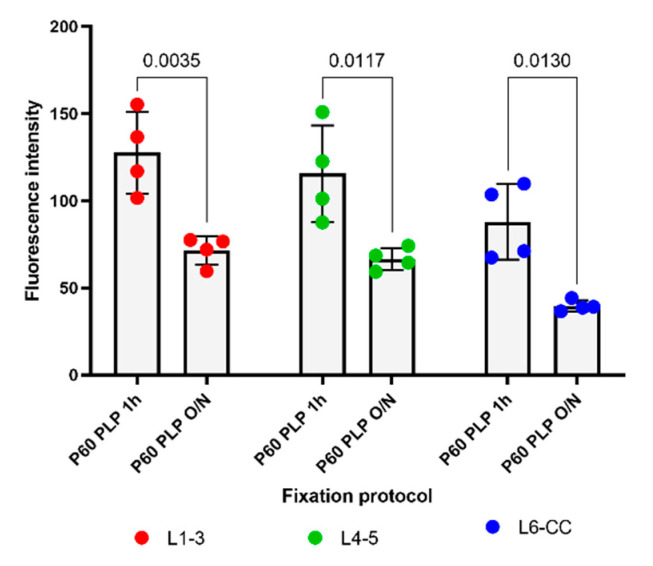
The effects of different lengths of PLP fixation on the intensity of PDGFRα immunofluorescence at P60 in different layers. Left: PDGFRα immunofluorescence intensity in L1–3 after PLP 1 h and PLP O/N. Middle: PDGFRα immunofluorescence intensity in L4–5 after PLP 1 h and PLP O/N. Right: PDGFRα immunofluorescence intensity in L6-CC after PLP 1 h and PLP O/N. A 2-way ANOVA followed by Tukey’s multiple comparisons test *n* = 4 for each group. Comparison among fixatives: F(1, 18) = 49.48, *p* < 0.0001. Comparison among layers: F(2, 18) = 8.791, *p* = 0.0022.

**Figure 5 cells-10-01302-f005:**
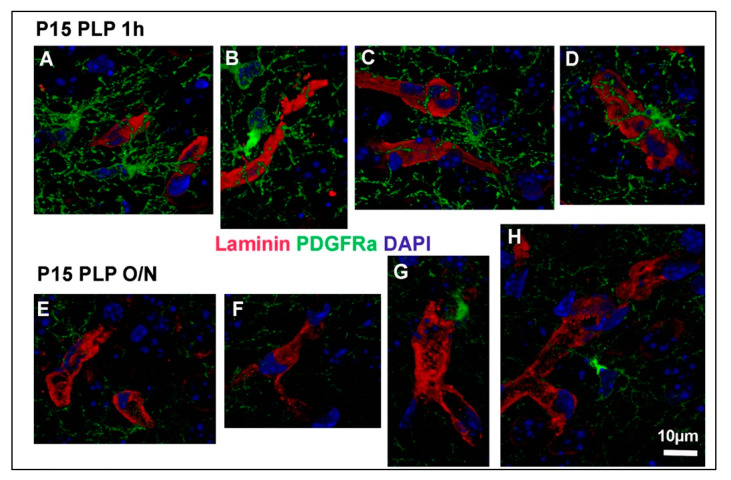
OPC protrusions contacting blood vessels. Three-dimensional reconstructions of OPCs contacting blood vessels are shown for P15 PLP 1 h (**A**–**D**) and P15 PLP O/N (**E**–**H**). Laminin-positive vessels are shown in red, PDGFRα-positive OPCs are shown in green, DAPI staining is shown in blue. Note that contacts are clearly visible in the short fixation protocol but hard to detect in the long fixation protocol. Regions of interest were taken from 40 z-stacks reconstructed with the Leica LAS X software for 3D reconstruction. Scale bar represents 10 µm.

**Figure 6 cells-10-01302-f006:**
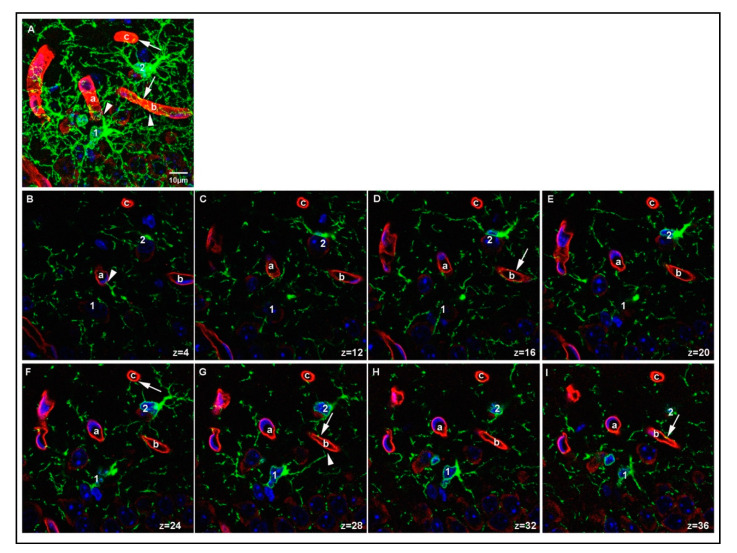
Details of OPC-blood vessel contacts. A region in the cortex of P15 PLP 1 h sample showing two OPCs (labeled 1 and 2 on the cell soma) contacting three different vessel segments (labeled a–c). A maximum projection of 40 z-stacks is shown in (**A**). Single *z*-slices (0.35 μm depth) from the stack showing individual protrusions contacting the surface of the vessel segments are shown in (**B**–**I**). Arrowheads indicate contacts made by OPC1. Arrows indicate contacts made by OPC2. Laminin-positive vessels are shown in red, PDGFRα-positive OPCs are shown in green, DAPI staining is shown in blue, position within the z-stack is indicated for each image. Scale bar represents 10 μm.

**Figure 7 cells-10-01302-f007:**
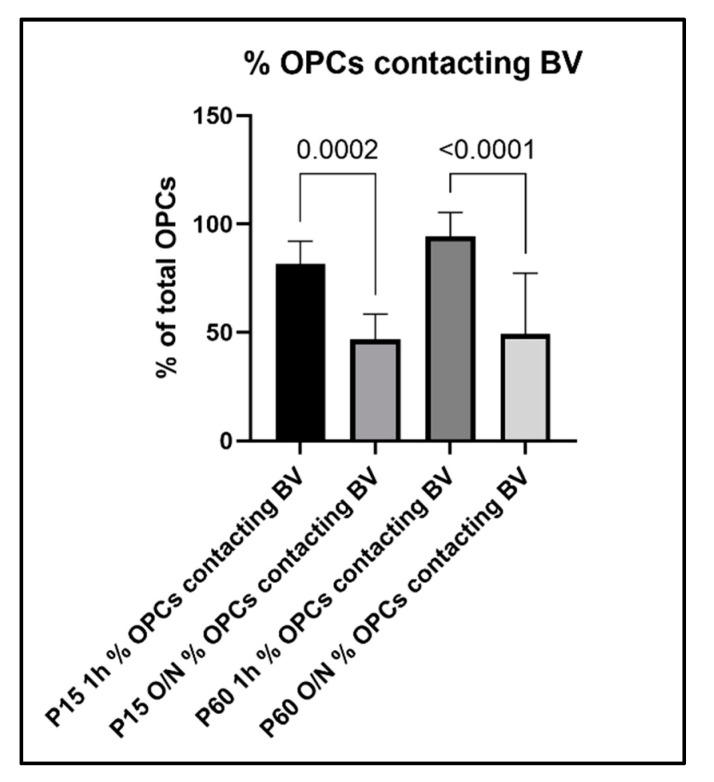
Percentage of OPCs that established contact to blood vessels in the different experimental groups.

**Figure 8 cells-10-01302-f008:**
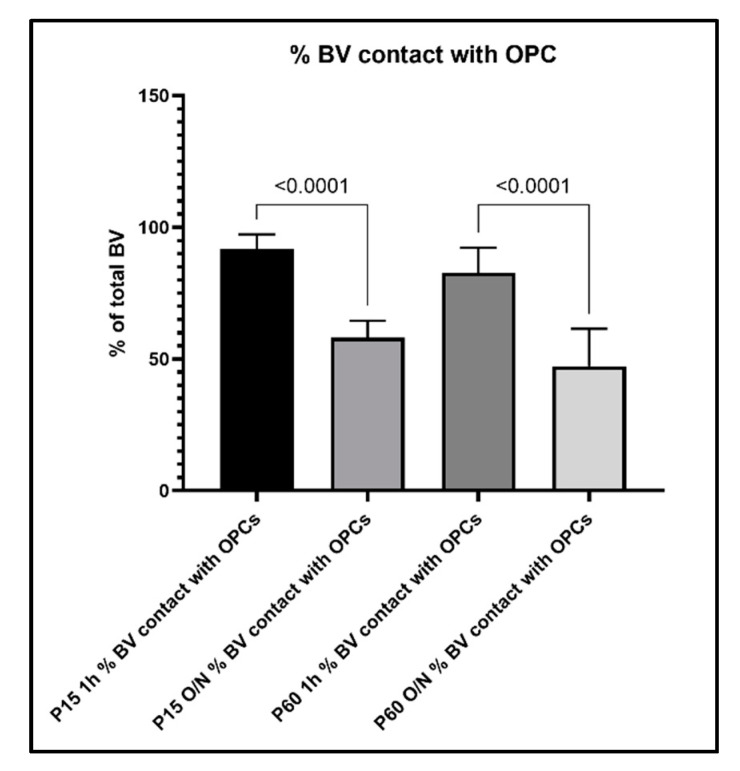
Percentage of blood vessel segments that were detected to have contacts with OPCs and their protrusions.

**Figure 9 cells-10-01302-f009:**
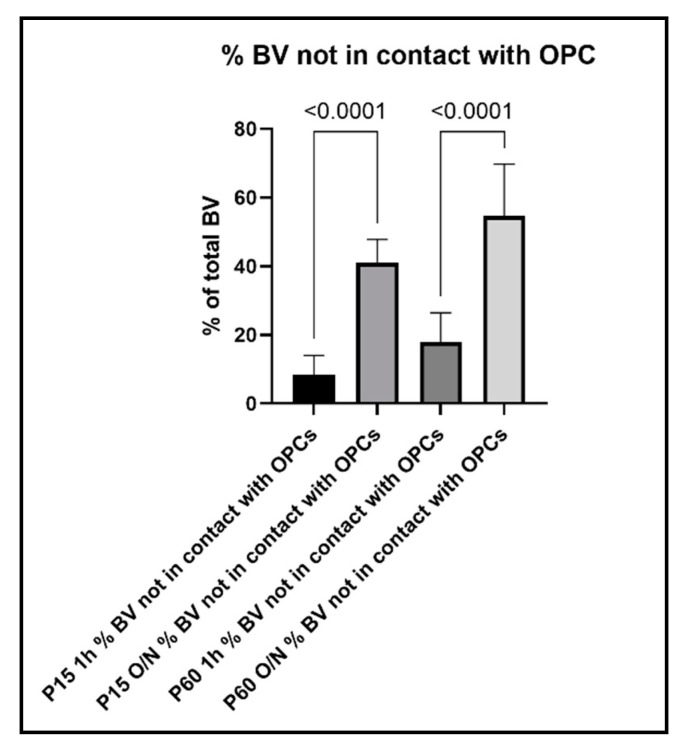
Percentage of blood vessels that are not in contact with OPCs and their protrusions.

**Figure 10 cells-10-01302-f010:**
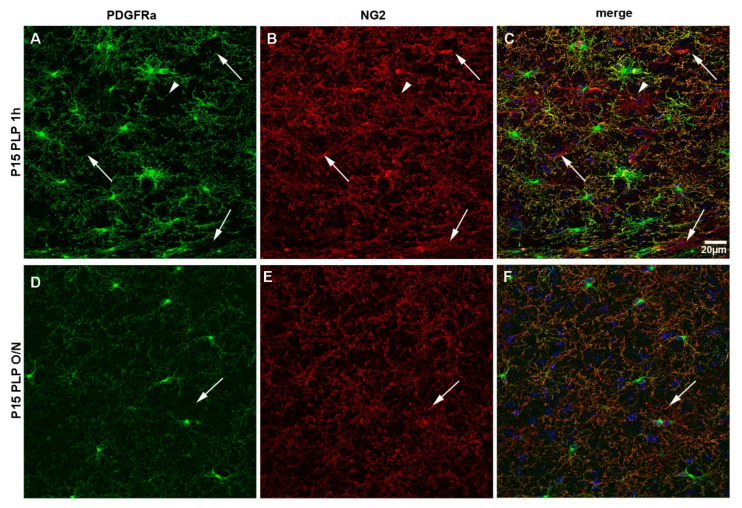
NG2 versus PDGFRα staining of OPCs. Comparison between NG2 and PDGFRα staining for the P15 PLP 1 h and the P15 PLP O/N groups. (**A**–**C**): Staining for PDGFRα (green) is more pronounced on the cell soma, but protrusions are clearly visible. Staining for NG2 (red) is less bright on the soma but more pronounced on protrusions. There are more protrusions visible with staining for NG2. Blood vessels are stained by NG2 but not by PDGFRα (pericytes, arrows). There are rare cells that are stained positively for NG2 but not PDGFRα (arrowhead). DAPI staining is shown in blue. (**D**–**F**): Overnight post-fixation with PLP decreases both PDGFRα and NG2 staining. PDGFRα, especially on distal protrusions, seems to be more vulnerable to long fixation compared to NG2. Scale bar represents 20 µm.

**Table 1 cells-10-01302-t001:** Experimental groups.

Age	Fixative for Perfusion	Fixative for Post-Fixation	Duration of Post-Fixation	Abbreviation
P15	4%PFA	4%PFA	1 h	P15 4%PFA 1 h
			overnight	P15 4%PFA O/N
	PLP	PLP	1 h	P15 PLP 1 h
			overnight	P15 PLP O/N
	0.1%GA/4%PFA	0.1%GA/4%PFA	1 h	P15 0.1%GA 1 h
	0.5%GA/4%PFA	0.5%GA/4%PFA	1 h	P15 0.5%GA 1 h
P60	PLP	PLP	1 h	P60 PLP 1 h
			overnight	P60 PLP O/N

## Data Availability

The data presented in this study are available at https://osf.io/g5vyu/files/ (accessed on 20 May 2021).

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
