# Peer review of "The Impact of Fixation on the Detection of Oligodendrocyte Precursor Cell Morphology and Vascular Associations"

_cells, 2021, doi:10.3390/cells10061302_

Round 1

Reviewer 1 Report

This is an interesting manuscript for 2 reasons:

1) It provides a better method for fixation and subsequent detection of OPC protrusions.

2) It provides indications that OPCs are connected to the brain vasculature.

There are however several points that need addressing:

1) PFA is introduced as the fixation that is used by many researchers (next to PFA-GA), and a shorter post-fixation is shown to be better than O/N fixation (which is generally used). The problem is however that a fixation with only PFA (4%) is not performed here, instead the PFA fixative also contains L-lysine and periodate, therefore called PLP. PLP is however more efficient in cross-linking than PFA on its own. It may therefore not be surprising that O/N PLP post-fixation gives over-fixation compared to 1-2 hrs. This observation may become much more relevant to other researchers if the results are compared to O/N PFA (4%) fixation (generally used).

3) Figure 4 shows very little difference between 1h and O/N fixation, while the pictures in Figure 3 suggest very large differences. Therefore, Figure 3 contains non-representative pictures.

4) Sentence 204 is confusing: “In P60 L1-5, the fluorescence intensity of PDGFR_labeling was about 1.2 to 1.3-fold 205 lower compared with that in P15 L1-5, but the difference was not significant between the 206 age groups (compare Figure 3A vs. 3B and Figure 3C vs. 3D)” It seems that instead a comparison should be made between Fig 1 (P15) and Figure 3 (p60). Also, the numbers on which these conclusion are based on are not presented.

5) The conclusion that: “our findings suggest that OPCs are an integral component of the neurovascular unit”, is not well supported by the data. Interactions between OPCs and blood vessels are shown, but this is not quantified. How many OPCs (%) have contact with a blood vessel? At this moment the findings could be rare observations, therefore having little meaning.

Minor:

  • The terms ‘two compartments’ in the abstract is not correct, it refers to OPCs, which are not a compartment.
  • Already in the introduction, the point should be made that PLP is used because of its efficient cross-linking of glycoproteins (like NG2 and PDGFR).
  • Line 105. A point is missing behind ‘measured’
  • Figure 7: 1 picture per conditions is sufficient (instead of 2)

Author Response

Dear Reviewer,

With this letter, we would like to answer the your comments. Our responses are indicated in italics below each comment.

  • PFA is introduced as the fixation that is used by many researchers (next to PFA-GA), and a shorter post-fixation is shown to be better than O/N fixation(which is generally used). The problem is however that a fixation with only PFA(4%) is not performed here, instead the PFA fixative also contains L-lysine and periodate, therefore called PLP. PLP is however more efficient in cross-linking than PFA on its own. It may therefore not be surprising that O/N PLP post-fixation gives over-fixation compared to 1-2 hrs. This observation may become much more relevant to other researchers if the results are compared to O/N PFA(4%) fixation (generally used).

We have added 2 groups for the P15 age group, where we used 4%PFA only fixation, with subsequent post-fixation for 1h or overnight, respectively. The results are included in Figures 1 and 2.

  • Figure 4 shows very little difference between 1h and O/N fixation, while the pictures in Figure 3 suggest very large differences. Therefore, Figure 3 contains non-representative pictures.

We noticed that the fluorescence intensity measurements for the P60 groups were not reflecting the images because it was picking up the background speckles that seemed higher at P60 than at P15. Therefore, we re-analyzed the fluorescence intensity for the P60 groups by using defined and equally sized squares around PDGFRa-positive cells, thus reducing the impact of the background signal on the values. This has been added to section 2.5 in the methods. We conclude that this greatly improved our analysis and include the new graph for Figure 4 in the revised manuscript.

  • Sentence 204 is confusing: “In P60 L1-5, the fluorescence intensity of PDGFR labeling was about 1.2 to 1.3-fold 205 lower compared with that in P15L1-5, but the difference was not significant between the age groups (compare Figure 3A vs. 3B and Figure 3C vs. 3D)” It seems that instead a comparison should be made between Fig 1 (P15) and Figure 3 (p60). Also, the numbers on which these conclusion are based on are not presented.

We have rewritten this section (3.2.2) to include the comparison between P15 and P60 and have clarified the ambiguities in the statements. 

  • The conclusion that: “our findings suggest that OPCs are an integral component of the neurovascular unit”, is not well supported by the data. Interactions between OPCs and blood vessels are shown, but this is not quantified. How many OPCs (%) have contact with a blood vessel? At this moment the findings could be rare observations, therefore having little meaning.

We include a new analysis showing the effects of fixation on the percentage of OPCs contacting blood vessels and the percentage of blood vessels being contacted by OPCs. It shows that under optimized fixation conditions, >80% of OPCs contact blood vessels, and most blood vessels (>80%) are contacted by OPC protrusions. Prolonged fixation significantly lowered these numbers, which underscore the importance of optimized fixation on detecting physiological phenomena.  

  • Minor: The terms ‘two compartments’ in the abstract is not correct, if refers to OPCs, which are not a compartment.

We changed the term to “elements”.

               Already in the introduction, the point should be made that PLP is used because of its efficient              cross-linking of glycoproteins (like NG2 and PDGFR).

               We have included this in the introduction.

               In Line 205, a point is missing behind ‘measured’

               We corrected this on line 116. 

               Figure 7: 1 picture per conditions is sufficient.

               Images were reduced for Figure 7, which is now Figure 10.

Reviewer 2 Report

Major comments:

The authors examined different fixative conditions to optimize the immunostaining for PDGFRa in OPCs. It is clearly shown that excessive fixation due to either too long duration of fixation process or the compositions of the fixatives impairs the antigenicity.

The description of the results is appropriate, and appropriate discussions have been developed. It seems to be a basic treatise that can provide useful information to OPC researchers.

However, the results of the analysis do not necessarily fully support the conclusions. That is; the immunostaining under hyperfixation conditions is concluded to impair the staining of protrusions rather than that of cell bodies; however, no quantitative results have been shown to directly support this. To draw the stated conclusion, it seems to be necessary to examine the effects of different fixation conditions on the number of identifiable OPCs and the specific fluorescence intensity of the part corresponding to cell body.

It may be necessary to reconsider the structure of the treatise? The descriptions from line 120 on page 3 to line 145 on page 5 partly overlap with those in "Materials and Methods", and do not include results. Therefore, it is necessary to reconsider whether it is desirable to incorporate this part into "Materials and Methods".

The direct contact of OPC process to laminin would be more directly visualized by some sorts of 3D demonstrations in Figure 6.

Perivascular cells are not clearly characterized in figure 7. Separately, single cell sequencing study (Vanlandewijck 2018) focusing on perivascular cells of normal mouse brain recently identified PDGFRa+/type 1 collagen+ perivascular fibroblasts as a cell population that are distinctive from NG2+ pericytes. Therefore, higher magnification views of perivascular cells may be necessary to be shown before concluding that “all PDGFRa positive cells at P15 stained positively for NG2 as well” at line 373 of page 13.

Minor comments:

“O/N” at line 195 of page 6 should be read “1h”.

“B-H” at line 273 of page 10 should be read “B-I”.

“It has recently been show that” at line 425 of page 14 should be read “It has recently been shown that”.

Author Response

Dear Reviewer,

With this letter, we would like to answer your comments. Our responses are indicated in italics below each comment.

  • The authors examined different fixative conditions to optimize the immunostaining for PDGFRa in OPCs. It is clearly shown that excessive fixation due to either too long duration of fixation process or the compositions of the fixatives impairs the antigenicity.

               The description of the results is appropriate, and appropriate discussions have been                developed. It seems to be a basic treatise that can provide useful information to OPC                researchers.

               However, the results of the analysis do not necessarily fully support the conclusions. That is; the immunostaining under hyperfixation conditions is concluded to impair the staining of                protrusions rather than that of cell bodies; however, no quantitative results have been                    shown to directly support this. To draw the stated conclusion, it seems to be necessary to           examine the effects of different fixation conditions on the number of identifiable OPCs and        the specific fluorescence intensity of the part corresponding to cell body.

               We included a new quantitative analysis, showing that cell numbers weren’t changed. It is            now the new supplementary Figure 4.

  • It may be necessary to reconsider the structure of the treatise? The descriptions from line 120 on page 3 to line 145 on page 5 partly overlap with those in "Materials and Methods", and do not include results. Therefore, it is necessary to reconsider whether it is desirable to incorporate this part into "Materials and Methods".

Although this is included in the Materials and Methods, we thought it would be important to elaborate this in the Results section to provide a rationale for the specific variables we have chosen to compare in this study.

  • The direct contact of OPC process to laminin would be more directly visualized by some sorts of 3D demonstrations in Figure 6.

The image panels in Figure 5 are 3D images, as indicated in the legend.

  • Perivascular cells are not clearly characterized in figure 7. Separately, single cell sequencing study (Vanlandewijck 2018) focusing on perivascular cells of normal mouse brain recently identified PDGFRa+/type 1 collagen+ perivascular fibroblasts as a cell population that are distinctive from NG2+ pericytes. Therefore, higher magnification views of perivascular cells may be necessary to be shown before concluding that “all PDGFRa positive cells at P15 stained positively for NG2 as well” at line 373 of page 13.

The objective of including this figure is to show that prolonged fixation diminishes immunodetection of different integral membrane proteins expressed on OPCs, and that the findings described in the other Figures are not due to a particular behavior of PDGFRa to fixation. The point raised by this reviewer on the identity of perivascular cells that express PDGFRa or NG2 is not the main focus of this study. We have modified the description of these data in the Results and Discussion sections to reflect the intended purpose. 

  • Minor comments:
  • “O/N” at line 195 of page 6 should be read “1h”.

Yes, it says 0.5% GA O/N in the Figure legend 2, line 191, which is not true, must be 0.5% GA 1h, it was changed.

  • “B-H” at line 273 of page 10 should be read “B-I”.

We changed it.

  • “It has recently been show that” at line 425 of page 14 should be read “It has recently been shown that”.

We corrected this in line 479.

Round 2

Reviewer 2 Report

non